# Electrochemical Sensors for Controlling Oxygen Content and Corrosion Processes in Lead-Bismuth Eutectic Coolant—State of the Art

**DOI:** 10.3390/s23020812

**Published:** 2023-01-10

**Authors:** Sergey N. Orlov, Nikita A. Bogachev, Andrey S. Mereshchenko, Alexandr A. Zmitrodan, Mikhail Yu. Skripkin

**Affiliations:** 1Institute of Chemistry, Saint-Petersburg State University, 7/9 Universitetskaya Emb., 199034 St. Petersburg, Russia; 2Federal State Unitary Enterprise “Alexandrov Research Institute of Technology”, 72, Koporskoe Shosse, 188540 Sosnovy Bor, Russia; 3Institute of Nuclear Industry, Peter the Great St. Petersburg Polytechnic University (SPbSU), 29, Polytechnicheskaya Street, 195251 St. Petersburg, Russia

**Keywords:** lead-bismuth eutectic coolant, structural materials, protective film, corrosion, oxygen activity sensors, impedance spectroscopy

## Abstract

Controlling oxygen content in the primary circuit of nuclear reactors is one of the key tasks needed to ensure the safe operation of nuclear power plants where lead-bismuth eutectic alloy (LBE) is used as a coolant. If the oxygen concentration is low, active corrosion of structural materials takes place; upon increase in oxygen content, slag accumulates due to the formation of lead oxide. The generally accepted method of measuring the oxygen content in LBE is currently potentiometry. The sensors for measuring oxygen activity (electrochemical oxygen sensors) are galvanic cells with two electrodes (lead-bismuth coolant serves as working electrode) separated by a solid electrolyte. Control of corrosion and slag accumulation processes in circuits exploring LBE as a coolant is also based on data obtained by electrochemical oxygen sensors. The disadvantages of this approach are the low efficiency and low sensitivity of control. The alternative, Impedance Spectroscopy (EIS) Sensors, are proposed for Real-Time Corrosion Monitoring in LBE system. Currently their applicability in static LBE at temperatures up to 600 °C is shown.

## 1. Introduction

Fast neutron reactors with lead or lead-bismuth eutectic coolant are one of the six promising reactor technologies selected within the framework of the GIF (Generation IV International Forum) international project as the basis for nuclear power of the future [1]. The advantages of lead-bismuth eutectic coolant are a relatively low melting point (125 °C), a high boiling point (above 1700 °C), low chemical activity compared to sodium and sodium-potassium coolant. All these factors determine a fairly wide range of nuclear power plants with lead-bismuth eutectic coolant currently being developed in Russia [2], USA [3], China [4,5] and Japan [6]. At the same time, the lead-bismuth eutectic coolant is highly corrosion-aggressive in relation to the structural materials of the first circuit [7] because of the significant solubility of structural materials components in lead.

The structural materials of the primary circuit are protected from the destruction by oxide films formed on their surface [8]. The structure and protective properties of these films depend on the chemical composition of structural materials, flow rate and temperature of the LBE and, primarily, on the oxygen content in the coolant [9]. Another problem that accompanies the exploration of reactors with lead-bismuth eutectic coolant is slag accumulation due to oxidation of the coolant if oxygen content in the circuit is too high. Insufficient attention that was paid to this issue at the early stage of operation led to a serious radiation accident with the melting of the core on the K-27 nuclear submarine of Project 645 [10].

Thus, an adequate choice of operating ranges and effective control of the oxygen content in the primary circuit is the basis for trouble-free operation of nuclear reactors where lead-bismuth eutectic coolant is explored. In addition, an important issue related to the aggressiveness of LBE in relation to structural materials is the organization of corrosion monitoring.

The aim of this review is to analyze existing approaches to the measurement of oxygen content and control of corrosion and slag accumulation processes in LBE. Special attention is paid to the use of oxygen activity sensors and alternative electrochemical methods for monitoring corrosion processes and slag accumulation in circuits with LBE. An additional task of the review is to familiarize the international scientific community with the results of the work of Russian scientists, currently published mainly in Russian.

## 2. Discussion

### 2.1. The Operating Range of Oxygen Concentrations

As was mentioned above, the protective properties of films formed upon oxygen inhibition of corrosion are determined by the conditions of their formation—the temperature and flow rate of the LBE, the temperature gradient in the circuit and the concentration of oxygen in the coolant. Depending on the oxygen content in the coolant, the conditions in the circuit can be divided into three groups [9]:(1)High oxygen content: from 1 × 10^−6^% to 1 × 10^−3^% by weight;(2)Average oxygen content: from 1 × 10^−6^%–1 × 10^−7^% by weight;(3)Low oxygen content: ≤1 × 10^−7^%.

If the oxygen content in the circuit is low, the structural materials are actively dissolved. A similar situation is also observed at temperatures so high that the resulting oxide layer cannot prevent the dissolution of structural materials [11]. At an average oxygen concentration, an oxide film forms on the surface of structural materials. At the same time, oxygen reacts with the dissolved components of steel. The resulting insoluble oxides of corrosion products are deposited on the surface of the steel, as a result of which a single-layer oxide film of ferro-chromium spinel (Fe_2-x_Cr_x_)_2_O_4_ is formed on the surface [11].

At a high oxygen concentration, the oxidation of iron atoms diffused outward from structural material takes place on the oxide film/LBE interface, resulting in the formation of an external oxide layer of magnetite. As not all oxygen is bound in this reaction, the remaining molecules migrate through the oxide layer and react with both iron and chromium under the film, increasing the thickness of the ferro-chromium spinel layer. As a result, a two-layer oxide film is formed on the surface of the structural material, in which only the inner layer has protective functions [12,13].

The processes of corrosion and oxidation of structural materials in the flow of liquid metal coolant differ from the case of static LBE. Klok [14] has shown that the corrosion rate is proportional to *v*^0.6−0.8^, where *v* is the linear velocity of LBE. The increase in the corrosion rate with an increase in the flow rate of the coolant is explained, firstly, by damage of the oxide layer due to erosion and, secondly, by accelerating the diffusion of corrosion products from the surface of the structural material into the volume of the coolant. The effect of the flow velocity on the structure of the oxide film is demonstrated by the results of Wiesenburg and his coauthors [15]. The corrosion resistance of martensitic steel T91 was analyzed. It was shown that, at an oxygen content of 10^−8^–10^−10^%, a coolant temperature of 450 °C, a flow rate of 1 m/s for 2000 h of the experiment and the penetration of LBE into the steel to a depth of 10 microns occurred. At an oxygen content of 10^−8^%, a coolant temperature of 300 °C and a flow rate of 1 m/s for 10,000 h of the experiment, neither the formation of an oxide film nor the dissolution of steel was observed. When the temperature of the coolant rose to 450 °C under similar conditions, a thin layer of oxide film was formed on T91 steel, preventing its corrosion. At an oxygen content of 10^−6^%, a coolant temperature of 480 °C, a flow velocity of 1.3 m/s and two-layer (magnetite and ferro-chromium spinel) film with a thickness of 28 microns was formed on steel during 6600 h of the experiment. An increase in temperature up to 550 °C under similar conditions led to an increase in the film thickness up to 45 microns. With an increase in the flow rate up to 2 m/s (oxygen content 10^−6^%, coolant temperature 550 °C) for 15,000 h of the experiment, a single-layer spinel film with a thickness of 45 microns was formed on T91 steel. Additionally, one of the experiments conducted in the LBE stream (1 m/s), at an oxygen concentration of 10^−8^% on the LINCE loop (Madrid), should be noted [16]—when the sample was held for 5000 h, the dissolution of T91 steel (up to 300 microns) was observed, although neither for samples with an exposure of 2000 h, nor for samples with an exposure of 10,000, such a pattern was not observed. The increased corrosion damage in the considered experiment was explained by the turbulent flow of LBE, which confirmed the influence of the characteristics of the coolant flow on the state of structural materials.

The areas of lead oxides and protective films formation depending on the temperature and oxygen concentration in the LBE [13,17,18] are shown in Figure 1.

As one can see from the data presented in Figure 1, if the temperature of lead-bismuth eutectic coolant was above 350 °C and oxygen concentration was above (7–8) × 10^−6^ wt %, lead monooxide was formed in the circuit. The optimal oxygen concentration values, which ensured the formation of strong protective films and at the same time prevented the formation of slag, are in a narrow range with (1–4) × 10^−6^ wt % width [19]. Currently, attempts are being made to expand this range by modifying the composition of structural materials and applying thin coatings enriched with elements with a high affinity for oxygen to their surface. It was shown by Ding et al. [20] that the energies of the element-oxygen bond in the LBE medium for the pairs Fe–O, Cr–O, Ni–O, Al–O and Si–O had the following values: −0.93; −1.26; −0.56; −1.45 and −1.75 eV, respectively. Thus, silicon, chromium and aluminum formed more stable protective oxide films compared to iron on the surface of structural materials. Aluminum is currently the most widely used material when applying protective coatings.

All the principal industrial methods of surface films formation were used for coating structural materials. Among them were chemical and physical deposition from the gas phase (batch cementing, pulsed laser spraying, vacuum arc technology, thermal spraying), mechanical milling and so on [21,22,23,24,25,26,27,28]. The main problem of the application of these coatings to inhibit corrosion of structural materials in the LBE environment was ensuring their sufficient mechanical strength and adhesion on the surface of the main alloy [27].

The method of high-current pulsed electron beams of microsecond range [29,30,31] is a promising approach to create protective coatings on the surface of structural materials in nuclear reactors with LBE. When the electron beam acted on the preliminary layer of aluminum, its melting with the structural material took place. As a result, an intermetallic phase of the Al_x_Fe_y_ composition with a depth of 20–30 microns formed on the surface. The mixing process had a vortex character. Because of the oxidation of the intermetallic phase in LBE, an extremely dense ultrathin Al_2_O_3_ oxide film formed on the surface of the material, which persisted during the testing process. This film completely prevented liquid metal corrosion at an oxygen content of 10^−8^–10^−7^ wt% and allowed the expansion of the oxygen range of normal operation of reactors with heavy liquid metal coolants [32].

Thus, maintaining the oxygen content at a given level in the LBE ensured the corrosion resistance of structural materials that determines the actuality of the monitoring of the oxygen concentration in the coolant.

### 2.2. Measurement of Oxygen in a Lead-Bismuth Eutectic Coolant

One of the most common ways to measure oxygen concentration in liquids is the electrochemical one [33]. The conventional electrochemical sensors to measure oxygen activity in a lead-bismuth coolant are a galvanic cell with two electrodes separated by a solid electrolyte. In sensors utilized at nuclear plants in Russia, the reference electrode is a saturated oxygen solution (with a constant oxygen activity equal to 1) in metals with a low melting point. Usually [34,35,36,37,38,39], Bi-Bi_2_O_3_ system (melting point 544 K) is used for the reference electrode. The operating temperature range of the oxygen sensor can be expanded by substitution of this system on In-In_2_O_3_ [40,41] (melting point 433 K) either Sn-SnO_2_ [42,43] (melting point 505 K) systems.

Alternative options for reference electrodes are:−Air/metal (Pt) [43,44] or perovskite oxide e.g., Lanthanum Strontium Manganite (LSM) and Lanthanum Strontium Manganite-Gadolinium Doped Ceria (LSM-GDC) [45,46];−Solid metal/metal oxide mixtures e.g., Fe/Fe_3_O_4_ [47,48,49,50], Cu/Cu_2_O [51].

The main disadvantage of the Pt-air reference electrode is the high minimum temperature of the readings—about 400 °C. At lower temperatures, especially in the area of low oxygen concentrations, a significant measurement error is possible [52,53].

To date, the possibility of using perovskites for high-temperature sensors has been proven [54]. At the same time, a significant disadvantage of all air electrodes is the possibility of contact of the coolant with the external atmosphere when the insulating layer of the electrode is damaged [44].

Liquid metal/metal oxide reference electrodes might cause the mechanical failure of the solid electrolyte once the melting point of the metal phase is crossed e.g., indium melts at 157 °C and LBE melts at 125 °C [44]. At the same time, an important advantage of the liquid bismuth/bismuth oxide reference electrode is the refinement of its technology and extensive experience in application of the sensors based on it both in reactors with static and with circulated coolant at numerous nuclear industry enterprises in Russia (JSC “NIKIET”, Moscow. CRISM “Prometey”, Saint-Petersburg, OKB “Giropress” JSC, Podolsk etc.) and abroad (e.g., ENEA center at Brazimone, Italy) [49].

As a working electrode, a lead-bismuth eutectic coolant acts, the oxygen activity in which is less than or equal to 1. Solid electrolytes separating the working and reference electrode in the sensors are solid solutions of the composition MeO_x_*Me’O_y_. The solutions are based on refractory oxides ZrO_2_, HfO_2_, ThO_2_, stabilized by the addition of CaO, Y_2_O_3_ and Sc_2_O_3_ [50,55]. These oxide additives provide ionic conductivity and mechanical strength of electrolytes due to stabilization of the cubic crystal lattice similar to fluorite. The highest ionic conductivity of electrolytes is achieved at concentrations of stabilizing additives of 7–15 mol % [50].

These oxides demonstrate n-type conductivity, which is preserved in solid solutions. As a result, solid electrolytes in oxygen sensors have mixed ion-electron conductivity. To account for this fact, the average transfer numbers were used—ion *t_i_* and electron *t_e_* [50]. These numbers are equal to the fractions of ionic and electronic conductivity, respectively.
(1)$$t_{i}+t_{e}\;=1$$

The ionic transfer numbers of the oxygen sensor electrolyte were determined experimentally. It is considered to be a conditioned sensor if *t_i_* is in the range 0.95–1.

When the sensor was operating, the ionization reaction of oxygen atoms proceeded on the reference electrode, followed by the transfer of anions through the electrolyte to the working electrode, where the anions were discharged [56]. It was assumed that oxygen dissolved in a liquid metal and in a reference, electrode was in atomic form [57] and was subject to the Sieverts law that allows the prediction of its solubility in metals.

Oxygen activity in LBE can be determined from the Nernst equation:(2)$$E_{T}=\frac{RT}{2F}\ln\left(\frac{a_{ref}}{a_{LBE}}\right)$$
where *E_T_* is the calculated EMF of the galvanic cell, V; *R* is the universal gas constant equal to 8.31 J/(mol × K); *F* is the Faraday number equal to 96,500 C/mol; *T* is the temperature of the coolant in the sensor area, K; *a_ref_*—thermodynamic activity of oxygen in the reference electrode; *a_LBE_*—thermodynamic activity of oxygen in LBE.

The calculated EMF of a galvanic cell correlates with the measured one as follows:E_o_ = *t_i_*∙E_T_(3)

For a sensor with a bismuth reference electrode in LBE, the relationship between the value of the measured EMF and activity of oxygen can be expressed in the form [58]:E_o_ = *t_i_*∙(0.088 − 9.91∙10^−5^T(0.18 + lg*a_LBE_*)),(4)
where thermodynamic activity of oxygen in LBE can be related to its concentration by the simple equation:*a_LBE_* = C*_LBE_*/C_s_(5)
where C*_LBE_* is the oxygen concentration in LBE, mass%; C_s_ is the solubility of oxygen in LBE, mass%. The solubility of oxygen in LBE can be calculated according to the equation:C_s_ = A − B/T(6)
where A and B are empirical coefficients. The values of parameters A and B obtained by various authors were summarized by Askhadullin and his coauthors and can be found in [49]. But Pityk [56] has mentioned that the use of the abovementioned equation to calculate the limit solubility of oxygen in LBE is not metrologically justified, and, therefore, the oxygen concentration values obtained using this equation can only be used as illustrative and have no meaning as physical quantities.

Another problem arising upon use of thermodynamic activity sensors in LBE was the polarization of the electrodes. The following polarization factors that changed the electrode potential can be implemented on the working electrode of the sensor [50]:−Oxides deposition on the surface of the solid electrolyte;−adsorption of metallic impurities on the electrolyte surface;−formation of microcracks;−reduction of the resistance of the interelectrode insulator;−the presence of open micropores on the surface of the electrolyte.

Polarizing factors increased the ionic current in the sensor circuit and the oxygen activity on the electrolyte surface, which led to an increase in the electrode potential by the polarization value *η*. Therefore, the measured EMF of the oxygen sensor was reduced by the same value:E = E_0_ − *η*.(7)

To account for the magnitude of polarization Musikhin [50] calculated the corresponding functions that allowed the modeling and predicting of the polarization processes on the working electrode.

Oxygen activity sensors are designed to work in a corrosive environment with additional exposure to radiation and neutron flux, which can cause additional damage to sensor materials [59]. Okubo and co-authors [60,61] have studied effect of the irradiation of a solid electrolyte—zirconium dioxide stabilized by the addition of yttrium oxide—by gamma radiation and electrons with an energy of 100 keV. They showed that bend strength was not affected by the irradiation even at high dose of 10 mGy. At the same time, for partially stabilized zirconia with yttria content of 3% by weight, a partial transition of the monoclinic phase to the tetragonal phase was observed.

Despite the harsh operating conditions, the lifetime (service life) of sensors in LBE is 4000 h or more [62]. To control possible sensor failures, it was proposed to install at least three DACs in the circuit [63]. In this case, before measuring the oxygen activity in the coolant, it was possible to check the operability of the sensors. This test consisted in measuring the potential difference between the reference electrodes of a pair of solid-electrolyte sensors. If this difference was constant within the absolute measurement error, these sensors were operational. If this condition was not met, then by sequentially sorting through pairs of sensors, an inoperable sensor was detected and excluded from further consideration.

To summarize the results, the measurement of oxygen content in the coolant was currently carried out by the potentiometric method; the hardware aspects of the method were well developed. At the same time, questions were raised about the relationship between the values obtained by means of sensors and the actual oxygen concentration in the coolant, as well as confirmation of the operability of sensors installed in the flow circuits with LBE.

### 2.3. Control of Corrosion and Slag Formation Processes in LBE Using Oxygen Activity Sensors

Diagnostics of the state of structural materials and protective coatings on their surfaces during the operation of nuclear power plants with LBE in Russia was carried out during the ejection of gas mixtures in isothermal mode at 300–330 °C [64]. Two variants of the introduced gas mixtures were provided: He + O_2_ and He + H_2_O(g) + N_2_ + H_2_.

When the first composition was used, a change in the thermodynamic activity of oxygen was considered an indicator. In the case of significant amounts of non-oxidized corrosion products of structural materials present in the coolant (liquid metal corrosion) or violation of protective coatings on the surface of the circuit takes place, the rate of deoxidation of the coolant increased. On the contrary, in the presence of significant amounts of slags (oxide compounds) in the circuit, the rate of deoxidation decreased. If a reducing gas mixture was used in diagnostic mode, the rate of increase in the concentration of hydrogen in the protective gas was monitored. In the presence of oxide slags in the coolant, the rate of hydrogen incoming decreased, in the presence of non-oxidized corrosion products, it increased. In fact, when this approach was used, the control was rather qualitative and did not provide a relationship with the quantitative values of the content of corrosion products and slag in the coolant.

Diagnostics of the state of the surfaces of structural materials of promising nuclear power plants with LBE was proposed to be carried out on the basis of monitoring the amount of dissolved oxygen supplied from the mass transfer apparatus [65]. An increase in this indicator reflected the intensification of corrosion processes, a decrease is the evidence of the accumulation of slags or the appearance of an additional source of oxygen. Establishing the relationship of these changes with the rates of corrosion processes and, moreover, the localization of corrosion sites, the determination of corroding equipment was practically impossible.

To increase the informativeness of the control of corrosion and slag accumulation processes based on the data of the oxygen activity sensors, Ivanov and Salaev have analyzed the relationship between the kind of temperature dependences of oxygen activity and the content of chemically active impurities in the coolant [66,67]. The deviation from the linear form of the dependence of the readings of the oxygen activity sensor on the temperature of the coolant takes place as a result of a change in the physico-chemical state of the coolant associated with the appearance of impurities in it. The researchers supposed that, in the future, a relationship will be developed that quantitatively links the oxygen activity sensor readings obtained in the modes of thermal cycling of the coolant with the content of impurities in the coolant, but at the moment this work has not been completed.

To summarize the results, the development of a quantitative relationship between the indications of oxygen activity sensors and corrosion processes in the circuit with LBE is a complex and non-trivial problem, the solution of which has not yet been obtained.

### 2.4. Impedance Spectroscopy Application to Monitor Corrosion Processes in LBE

At the moment, the Electrochemical Impedance Spectroscopy method is widely used to monitor the corrosion processes in various environments [68,69,70,71,72]. This method is used to study the structure of corrosion deposits and the mechanism of their formation [73], the resistance of coatings on the surface of materials [74] and the effectiveness of corrosion inhibitors [75]. The advantage of the Impedance Spectroscopy method is its ability to track changes in the state of the metal/liquid metal interface, which is extremely important in the case of using oxide films to inhibit corrosion processes in LBE.

Electrochemical impedance spectroscopy is based on monitoring the response of an electrochemical system to a sinusoidal disturbance of small amplitude (several mV) in a wide frequency range (10^7^ to 10^−4^ Hz). Quantitative analysis of the frequency dependence of the impedance on the basis of the selected equivalent circuit makes it possible to interpret its elements in accordance with the physicochemical nature of the processes occurring on the electrodes [76,77].

Initially, the application of this method to the study of oxide films in LBE was proposed by Lillard and Stubbins in 2004–2008 [78,79,80]. A small sinusoidal voltage excitation was measured between the sample and reference electrode in a three-electrode configuration, where the current from the working electrode passed through a counter electrode and the impedance was measured as a function of frequency. In the first set of the experiments, the working electrode represented a metal rod in a ceramic tube [81] and impedance spectroscopy probe was formed by pushing a stainless-steel welding rod down the center of a ceramic tube, until the tip of the rod penetrated very slightly from the end of the tube. A steel reference electrode was also placed in the LBE. The difference between the impedances measured from the steel with and without the oxide layers indicated the existence of the oxide layers in the first case, and the magnitude and/or kind of difference yielded information about processes occurring on the electrodes [77]. This device produced reasonable results only at temperatures up to 200 °C or 300 °C, which was not enough to organize real-time corrosion monitoring in the circuits of nuclear reactors with LBE.

To solve this problem the modification of working electrode was suggested [81,82]. The modified electrode consisted of a 410 stainless-steel disk, completely coated with alumina except for two small areas on the back side, for electrical connections, and one small area on the front side, acting as the active area to be exposed to the LBE. This active area was a circle of 0.25-inch diameter. This disk was then pushed onto the mostly closed bottom of a stainless-steel cylinder; the bottom had a 0.375-inch-diameter hole drilled through it to allow the active area of the disk to communicate with the outside of the cylinder.

This improvement of working electrode made it possible to raise the operating range of the impedance spectroscopy method in SVT to a temperature of 600 °C. Kondo [83,84] has shown that the application of this method in a lead coolant is possible up to a temperature of 500 °C.

A simplified Randles electrical equivalent circuit (EEC), without the Warburg diffusion element [84], was used for equivalent circuit fitting in liquid metal, as shown in Figure 2. For equivalent-circuit fitting, a parallel combination of a resistor and constant-phase element (CPE) was used. The resistor and CPE in this circuit have been assigned to the resistance and capacitance of the oxide layer.

CPE Impedance *Z_CPE_* [78] is equal to:(8)$$Z_{CPE}=\frac{1}{A{\left(iw\right)}^{n}}$$
where *w*—frequency, Hz; *A*—frequency dependent parameter; *n*—mathematical constant without obvious physical sense.

The equivalent capacitance of the surface oxide film can be calculated as
(9)$$C=R_{oxide}^{\frac{1-n}{n}}A^{\frac{1}{n}}$$
where *R_oxide_*—polarization resistance of the surface oxide film formed on sample, Ohm.

All the parameters could be acquired from the fitting of the Nyquist plot, based on the experimental data. The Nyquist plot is a complex ohmic plane, impedance real component Z′ was deposited along the *x* axis, and the imaginary component of resistance Z′′ was deposited along the *y* axis
(10)$$Z=Z^{\prime }-iZ^{\prime \prime }$$
(11)$$Z^{\prime }=Z_{0}\cos\varphi$$
(12)$$Z^{\prime \prime }=Z_{0}\sin\varphi$$
where *Z*_0_ is voluminous impedance of the sample and φ is phase angle.

When constructing the impedance spectrum in Nyquist coordinates, to obtain each point of the curve, the values *Z*′ and *Z*″ are calculated from the data array (*ω*, *Z*_0_, *φ*) or the vector *Z*_0_ is placed on the plane at an angle φ to the *x* axis and the position of its endpoint is fixed (the coordinate grid must be square).

The example of Nyquist plot taken from [85] together with fitted data is shown at Figure 3 below. Measured and fitted impedance agree one another fairly well.

From equivalent capacitance the thickness of oxide film can be calculated as follows:(13)$$d=\frac{\varepsilon \varepsilon_{0}s}{C}$$
where *d* is oxide layer thickness, μm; *ε*—absolute permittivity, (F/m); *ε*_0_—vacuum permittivity, (F/m); s—the electrode area (surface exposure area of the specimen in LBE), cm^2^.

Qiu et al. [86] considered the impedance properties of iron oxide films in LBE. They showed that the impedance value was proportional to the thickness of the film. When the thickness of the iron oxide film was less than 50 nm, the impedance value was insignificant. With an increase in the thickness of the oxide film to 200 nm, the impedance value increased to 1 kΩ cm^2^. Disruption of the oxide via scratching led to an immediate loss of impedance due to short circuiting. This fact testifies to the possibility of impedance spectroscopy sensors application to detect damage to the oxide film on structural materials in LBE in real time.

The next step towards the introduction of impedance spectroscopy into the corrosion monitoring system of structural materials of the first circuit of the nuclear reactors with LBE coolant is the testing of this method in LBE streams at circulation stands that allow us to draw the final conclusion about the applicability of this kind of sensors.

## 3. Conclusions

The elements that make up austenitic and martensitic steels were effectively dissolved in lead-bismuth coolant. Oxide films protected the structural materials of the primary circuit from destruction.To preserve the protective functions of oxide films, a stable oxygen concentration was maintained in the primary coolant in a narrow range from 1 × 10^−6^% to 4 × 10^−6^% or up to 1 × 10^−8^–1 × 10^−7^% when using protective coatings made of aluminum oxide.Measurement of the oxygen content in the LBE was carried out by the method of potentiometry. At the moment, this method is fully developed, but does not exclude a number of metrological difficulties in interpretation and ensuring the reliability of measurement results.The use of oxygen activity sensors to monitor the processes of corrosion and slag formation in LBE was hampered by the lack of a quantitative relationship between the sensor readings and the content of impurities in the coolant.To obtain data on the state of protective oxide films (including thickness and mechanical damage) on structural materials in static LBE at temperatures up to 600 °C, impedance spectroscopy sensors can be used.The next stage for the introduction of impedance spectroscopy sensors into the corrosion monitoring system of the first circuit of promising nuclear reactors with LBE is to confirm their operability in flowing LBE loops.

## Figures and Tables

**Figure 1 sensors-23-00812-f001:**
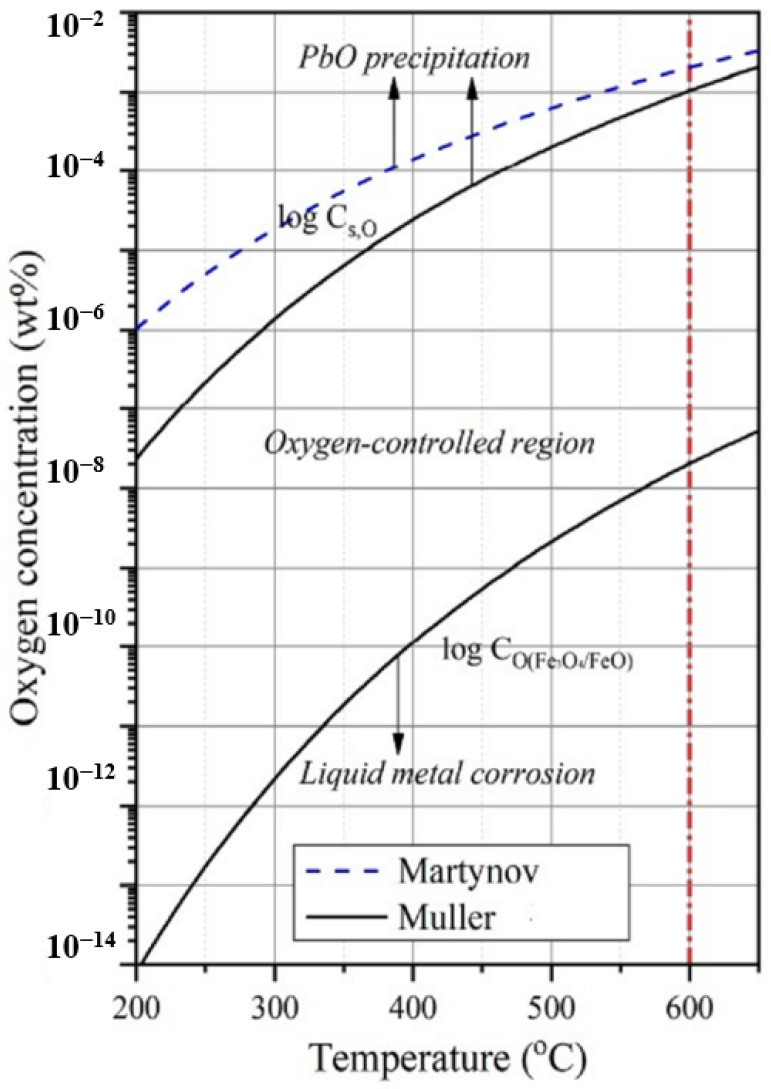
Conditions of lead oxide and protective films formation according to Martynov [17] and Müller [18]. The red line indicates the maximum core temperature for currently designed reactors with LBE [13].

**Figure 2 sensors-23-00812-f002:**
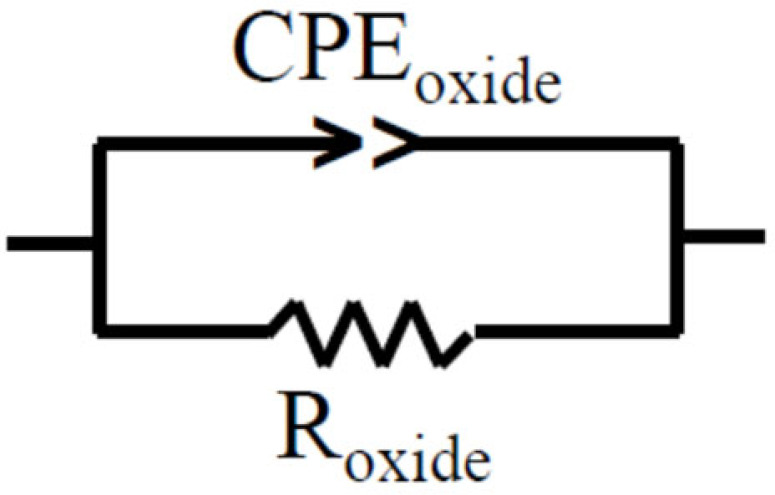
Electrical equivalent circuit used to fit impedance data.

**Figure 3 sensors-23-00812-f003:**
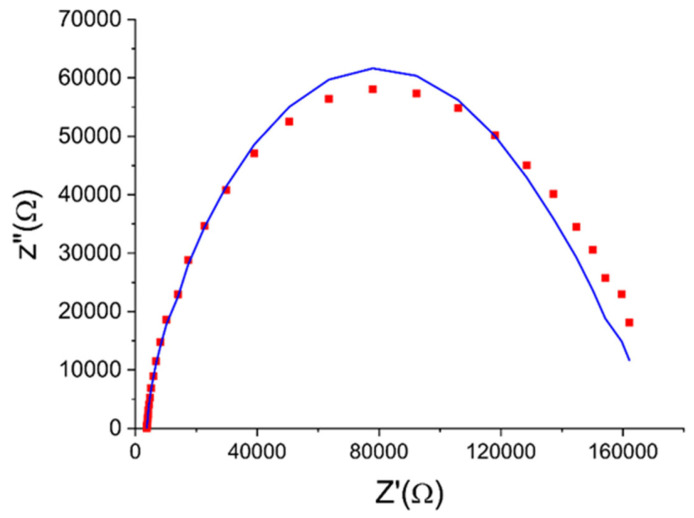
Niquist plot together with fitting parameters calculated. Squares—experimental data, solid line—fitted one.

## Data Availability

Not applicable.

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
