# Peer review of "Electrochemical Sensors for Controlling Oxygen Content and Corrosion Processes in Lead-Bismuth Eutectic Coolant—State of the Art"

_sensors, 2023, doi:10.3390/s23020812_

Round 1

Reviewer 1 Report

The paper reviews monitoring technologies to control oxygen content in the primary circuit of nuclear reactors. It is important topic to ensure the safe operation of nuclear power plants with lead-bismuth eutectic alloy (LBE) as a coolant.

Main focused region is approximately 10^-7 wt% oxygen because if the oxygen concentration is much lower, chemical corrosion of structural materials can take place. If it is much higher than the region, PbO precipitation can form in LBE.

This manuscript is well written in this current form. There are minor comments from this reviewer.

1. 2nd line in abstract, "were" could be an error in spelling.

2. The author wrote "lead oxides and protec-
tive films formation depending on the temperature and oxygen concentration in the LBE" with 14 and 15 references. Is the relation applicable in various flow conditions? Some papers report oxide film is removed in a flow condition and it results in a weight loss of structural materials.

3. It would be good if the authors add descriptions about life-time of each monitoring technique. In general, sensor used in a very high temperature is corroded and thus its life time is not long.

Author Response

Dear reviewer,

First of all, we would like to express our deep gratitude for the high appreciation of the work and valuable comments.

As to your comments, the next changes were introduced into the text:

  1. 1. 2nd line in abstract, "were" could be an error in spelling.

- Corrected

2. The authors wrote "lead oxides and protective films formation depending on the temperature and oxygen concentration in the LBE" with 14 and 15 references. Is the relation applicable in various flow conditions? Some papers report oxide film is removed in a flow condition and it results in a weight loss of structural materials.

- Yes, the processes of corrosion and oxidation of structural materials in the flow of liquid metal coolant differ from the case of static one. According to the literature data the corrosion rate is directly proportional V0,6-0,8, where V is the linear velocity of LBE. The increase in the corrosion rate with an increase in the flow rate of the coolant is explained, firstly, by the damage to the oxide layer due to erosion; secondly, by the acceleration of the diffusion of corrosion products from the surface of the structural material into the volume of the coolant. The discussion of this topic together with some examples is added to the text at the end of 2.1 paragraph.

  1. It would be good if the authors add descriptions about life-time of each monitoring technique. In general, sensor used in a very high temperature is corroded and thus its life time is not long.

The next data were added to the manuscript:

Operating time (service life) of oxygen activity sensors in LBE is 4000 hours or more. For impedance spectroscopy sensors, the resource characteristics have not yet been fully studied, since the application of this method for SVT is at an initial stage. Data on the effect of ionizing radiation on sensors has also been added to the text.

Yours sincerely,

Authors

Reviewer 2 Report

I have critically reviewed the manuscript entitled “Electrochemical sensors for control oxygen content and corro-sion processes in lead-bismuth eutectic coolant – state of the art”. The experiments conducted in the manuscript are very systematic. Results have also been discussed critically. This manuscript should be accepted but after undergoing major revisions. My comments are as follows:

1.     The first line of the abstract has got grammatical error, please correct it.

2.     The following research articles should be referred and cited in the introduction section of the manuscript for its technical soundness.

·       Neupane S, Subedi V, Thapa KK, Yadav RJ, Nakarmi KB, Gupta DK, Yadav AP. An alternative pH sensor: graphene oxide-based electrochemical sensor. Emergent Materials. 2022 Apr;5(2):509-17.

·       Narwade VN, Bogle KA, Kokol V. Hydrothermally synthesized hydroxyapatite cellulose composites thick films as ammonia gas sensor. Emergent Materials. 2022 Apr;5(2):445-54.

·       Al Mannai A, Haik Y, Elmel A, Qadri S, Saud KM. 3D SERS-based biosensor for the selective detection of circulating cancer-derived exosomes. Emergent Materials. 2021 Nov 18:1-3.

·       Sur UK, Santra C. Spectroscopy: a versatile sensing tool for cost-effective and rapid detection of novel coronavirus (COVID-19). Emergent Materials. 2022 Feb 28:1-2. On page no. 3, second last paragraph there is a presence of typos (too much use of “the”). Please remove it.

3.     How structural materials of the primary circuit are protected from the destruction by oxide films formed on their surface?

4.     Why high-current pulsed electron beams of microsecond range are an important technique?

5.     Why are air electrodes easy to manufacture? Discuss.

6.     Why with an increase in polarizing factor leads to a decrease in measured EMF of the sensor?

7.     How did disruption of the oxide via scratching led to an immediate loss of impedance due to short circuiting?

8.     Conclusion of the manuscript is incomplete, please include other points.

Author Response

Dear reviewer,

First of all, we would like to express our deep gratitude for the high appreciation of the work and valuable comments.

As to your comments, the next changes were introduced into the text:

  1. The first line of the abstract has got grammatical error, please correct it.

- Corrected.

  1. The following research articles should be referred and cited in the introduction section of the manuscript for its technical soundness.

Neupane S, Subedi V, Thapa KK, Yadav RJ, Nakarmi KB, Gupta DK, Yadav AP. An alternative pH sensor: graphene oxide-based electrochemical sensor. Emergent Materials. 2022 Apr;5(2):509-17.

Narwade VN, Bogle KA, Kokol V. Hydrothermally synthesized hydroxyapatite cellulose composites thick films as ammonia gas sensor. Emergent Materials. 2022 Apr;5(2):445-54.

Al Mannai A, Haik Y, Elmel A, Qadri S, Saud KM. 3D SERS-based biosensor for the selective detection of circulating cancer-derived exosomes. Emergent Materials. 2021 Nov 18:1-3.

Sur UK, Santra C. Spectroscopy: a versatile sensing tool for cost-effective and rapid detection of novel coronavirus (COVID-19). EmergentMaterials. 2022 Feb 28:1-2

- These articles are certainly very interesting, but they are far from the subject of this review: they do not concern either the nuclear industry or the determination of oxygen. From our point of view, quoting them seems to be superfluous.

  1. How structural materials of the primary circuit are protected from the destruction by oxide films formed on their surface?

- The oxides of the elements that make up the structural materials do not dissolve in LBE. As a result, their formation on the surface prevents the process of dissolution of the structural material itself.

  1. Why high-current pulsed electron beams of microsecond range are an important technique?

- When the electron beam acts on the preliminary layer of aluminum, its melting also occurs with the structural material. As a result, an intermetallic phase of the AlxFey composition with a depth of 20-30 microns is formed on the surface. The mixing process has a vortex character.

  1. Why are air electrodes easy to manufacture? Discuss.

- Yes, this statement is controversial, so we have removed it from the manuscriopt.

  1. Why with an increase in polarizing factor leads to a decrease in measured EMF of the sensor?

- Polarizing factors increase the ion current in the sensor circuit and the oxygen activity on the electrolyte surface, which leads to an increase in the electrode potential by the amount of polarization. Therefore, the measured EMF of the oxygen sensor is reduced by the same amount.

  1. Как механические повреждения оксида приводят к немедленной потере импеданса?

How did disruption of the oxide via scratching led to an immediate loss of impedance due to short circuiting?

  • The impedance value depends on the resistance and capacitance of the oxide film. When removing the film on a part of the sample surface – scratching – these indicators fall to zero and the impedance disappears.
  1. Conclusions are incomplete, some other points should be added.
  • We have re-written the conclusions.

Yours sincerely,

Authors.

Reviewer 3 Report

Comments on the paper:

The manuscript titled: "Electrochemical sensors for control oxygen content and corrosion processes in lead-bismuth eutectic coolant – state of the art" is very informing one.

 There are a few comments and questions.

1.      It is not clear what the new results of the manuscript are or only an overview of the previous results are given. The authors should clearly state what they results are.

2.      There are some typos: a) On page 2 at the end of the first paragraph "in lead" is duplicated; On page 3, it is written (7-8)*10-6 wt%, than it is written (1-4)·10-6 wt%, please write it in the same way ("*", or "·").

Author Response

Dear reviewer,

First of all, we would like to express our deep gratitude for the high appreciation of the work. As to your comments, the next changes were introduced into the text:

  1. It is not clear what the new results of the manuscript are or only an overview of the previous results are given. The authors should clearly state what they results are.

- This paper is an overview of the previous results only. We apologize for the mistake took place during submission process: that should be “review” and not “article”

  1. There are some typos: a) On page 2 at the end of the first paragraph "in lead" is duplicated; On page 3, it is written (7-8)*10-6 wt%, than it is written (1-4)·10-6 wt%, please write it in the same way ("*", or "·").

- Corrected

Yours sincerely,

Authors.

Reviewer 4 Report

REPORTS ON: sensors-2087738

The proposed manuscript is reasonably interesting. However, if a state-of-art is intended, it seems that some weaknesses are identified. For instance, less than 30% of articles/book published between 2018~2022 are cited. Also, from those cited papers, more than 6~7 (~10%) are self-citation. From these, conference and patent are also cited. These should be prevented. Some few articles in Russian language are also cited. This should also be prevented. It is considered that the content is “weak” when a study of art is proposed. Into both Introduction and discussion section, it is not clear the NOVELTY or at least the “real” CONTRIBUTION (in both scientific and technological aspects). Additionally, only one reference in SENSORS (MDPI) journal is cited. Considering the organization of the proposed manuscript, the schematic representation of a sensor system should be depicted. Also, there is a great number of Equations described, which could be prevented. Citing articles would be perfectly understood the equation discussion. When discussing the electrochemical aspect into the subsection 2.4, this is rather and poorly detailed. Schematic representation is absent and formulations are not mentioned. At least this section is confusing and no contribution is attained. Based on these weaknesses mentioned, in my frank opinion, the manuscript not deserves its publication.

Author Response

Dear reviewer,

First of all, we want to thank you for the attention paid to our paper and for the valuable comments. Below you will find our answers to your comments.  

The proposed manuscript is reasonably interesting. However, if a state-of-art is intended, it seems that some weaknesses are identified. For instance, less than 30% of articles/book published between 2018~2022 are cited.

  • We have added 12 more references published after 2018, including 5 ones published in Sensors.

Conference and patent are also cited. These should be prevented. Some few articles in Russian language are also cited. This should also be prevented.

- Familiarization of the English-speaking audience with the results of the work of the Russian scientific school, the only one with successful experience in the operation of reactors with LBE, was one of the objectives of this review. The specificity of Russian works in this field is the rarity of publications in international journals, most of the original research is presented only at conferences and in the form of patents. Therefore, we consider it justified to include conference materials, patents and Russian-language articles in the list of references.

- Considering the organization of the proposed manuscript, the schematic representation of a sensor system should be depicted. Also, there is a great number of Equations described, which could be prevented. Citing articles would be perfectly understood the equation discussion. When discussing the electrochemical aspect into the subsection 2.4, this is rather and poorly detailed. Schematic representation is absent and formulations are not mentioned. At least this section is confusing and no contribution is attained. 

- Text is modified according to these comments.

Yours sincerely,

Authors.

Reviewer 5 Report

The authors reported the electrochemical sensors for controlling oxygen content and corrosion processes in lead-bismuth eutectic coolant. This manuscript needs major revision and the authors must pay attention and extensive editing for the English language to be considered for publication. The following are some of my comments: -

 1-  The title has the word “control” should be changed to “controlling”.

2- Abstract is very long and not concise, this part should explain the work in a short paragraph.

3- The sentence “Control of oxygen content in the primary circuit of nuclear reactors is one of the key tasks to ensure the safe operation of nuclear power plants were lead-bismuth eutectic alloy (LBE) is uses as a coolant.” is too long and is not understood, why “were” here (maybe you mean “where”) and also “is uses” what does it mean?

4- The authors wrote 3 paragraphs in the Abstract to introduce their work and even the 4th and the last paragraph does not express the idea of their work. Authors are requested to rewrite the Abstract again concluding the main findings of their work.

5- The Introduction Part is fine but needs Extensive English revision. Also, the authors wrote in the last line in the Introduction “The aim of this review is to analyze modern approaches to the measurement of oxygen content and control of corrosion and sludge accumulation processes in LBE.”, is this work a Review or an Article? Because the type of manuscript is written “Article” in the top left of the first page of the manuscript. This manuscript cannot be published as an article because it does not have experimental results.

6- Take care of the citation of references in the text, like Qiu [65] should be Qiu et al. [65].

7- The authors cited only one paper published in Sensors (Reference 55), I believe there are many published papers in Sensors worth to be cited.

The authors should extensively revise the manuscript and make it as State of the Art paper.

Author Response

Dear reviewer,

First of all, we would like to express our deep gratitude for your valuable comments. The next changes were introduced into the text:

 1-  The title has the word “control” should be changed to “controlling”.

  • corrected

2- Abstract is very long and not concise, this part should explain the work in a short paragraph.

  • corrected

3- The sentence “Control of oxygen content in the primary circuit of nuclear reactors is one of the key tasks to ensure the safe operation of nuclear power plants were lead-bismuth eutectic alloy (LBE) is uses as a coolant.” is too long and is not understood, why “were” here (maybe you mean “where”) and also “is uses” what does it mean?

  • corrected

4- The authors wrote 3 paragraphs in the Abstract to introduce their work and even the 4th and the last paragraph does not express the idea of their work. Authors are requested to rewrite the Abstract again concluding the main findings of their work.

  • Abstract is re-written.

 5- The Introduction Part is fine but needs Extensive English revision. Also, the authors wrote in the last line in the Introduction “The aim of this review is to analyze modern approaches to the measurement of oxygen content and control of corrosion and sludge accumulation processes in LBE.”, is this work a Review or an Article? Because the type of manuscript is written “Article” in the top left of the first page of the manuscript. This manuscript cannot be published as an article because it does not have experimental results.

- This paper is certainly a review. We apologize for the mistake took place during submission process: that should be “review” and not “article”

6- Take care of the citation of references in the text, like Qiu [65] should be Qiu et al. [65].

  • corrected

 7- The authors cited only one paper published in Sensors (Reference 55), I believe there are many published papers in Sensors worth to be cited.

  • 5 more papers published in Sensors have been added to reference list.

Yours sincerely,

Authors.

Round 2

Reviewer 4 Report

Although some improvements are provided, some other weaknesses are still verified. For instance, when "trying" to discuss electrochemical aspects, mainly Nyquist plot, few references are correlated. Additionally, at least one example of Nyquist plot and their correlated impedance parameters should be demonstrated and discussed. This based on the fact that, if the electrochemical impedance spectroscopy is a potential method to be used, a better and more explained section should be provided. Besides, it is hardly suggested that a new sentence be included in order to clarify that EIS is a commonly technique used to evaluate both oxide film formation and corrosion behavior of distinctive materials into different corrosive condition. The follow references (chronological sequence) can be used and cited:

[AA] Duarte T, Meyer Y.A. Osório W.R. The Holes of Zn Phosphate and Hot Dip Galvanizing on Electrochemical Behaviors of Multicoatings on Steel Substrates. Metals 2022, 12(5): 863.

[BB] Y.A Meyer, RS Bonatti, AD Bortolozo, WR Osório. Electrochemical behavior and compressive strength of Al-Cu/xCu composites in NaCl solution. Journal of Solid State Electrochemistry 25 (2021) 1303-1317.

Finally, but no more less important, English written should meticulously revised and improved. For instance, into Conclusion section (yellow highlighted) the term "matrensitic steels" is erroneously written.

Author Response

First of all, we are deeply grateful to the reviewer for his extremely valuable advice, which allowed us to improve the article.

We have re-written the sentence at the beginning of 2.4 section to clarify the importance of EIS to monitor both oxide film formation and corrosion processes.  Also we have added several lines to show how to treat the Nyquist plot to get parameters necessary for further calculations.

    With best wishes,

    Yours sincerely,

    Authors

Reviewer 5 Report

The authors have made the necessary corrections and performed my suggestions as well as the manuscript has been greatly improved. I recommend accepting it for publication.

Author Response

We are very thankful to the reviewer for his comments.